# Family vulnerability scale: Evidence of content and internal structure validity

**Evelyn Lima de Souza**[1]*, **Ilana Eshriqui**[1], **Flávio Rebustini**[2], **Eliana Tiemi Masuda**[1], **Francisco Timbó de Paiva Neto**[1], **Ricardo Macedo Lima**[1], **Daiana Bonfim**[1]

**1** Hospital Israelita Albert Einstein, São Paulo, São Paulo, Brazil, **2** Department of Gerontology, School of Arts, Sciences and Humanities (EACH), University of São Paulo, São Paulo, São Paulo, Brazil

* evelyn.lima@einstein.br

**Data Availability Statement:** All relevant data are within the paper and its Supporting Information files.

## Abstract

### Introduction

Territory view based on families' vulnerability strata allows identifying different health needs that can guide healthcare at primary care scope. Despite the availability of tools designed to measure family vulnerability, there is still a need for substantial validity evidence, which limits the use of these tools in a country showing multiple socioeconomic and cultural realities, such as Brazil. The primary objective of this study is to develop and gather evidence on the validity of the Family Vulnerability Scale for Brazil, commonly referred to as EVFAM-BR (in Portuguese).

### Methods

Items were generated through exploratory qualitative study carried out by 123 health care professionals. The data collected supported the creation of 92 initial items, which were then evaluated by a panel of multi-regional and multi-disciplinary experts (n = 73) to calculate the Content Validity Ratio (CVR). This evaluation process resulted in a refined version of the scale, consisting of 38 items. Next, the scale was applied to 1,255 individuals to test the internal-structure validity by using the Exploratory Factor Analysis (EFA). Dimensionality was evaluated using Robust Parallel Analysis, and the model underwent cross-validation to determine the final version of EVFAM-BR.

### Results

This final version consists of 14 items that are categorized into four dimensions, accounting for an explained variance of 79.02%. All indicators were within adequate and satisfactory limits, without any cross-loading or Heywood Case issues. Reliability indices also reached adequate levels (α = 0.71; ω = 0.70; glb = 0.83 and ORION ranging from 0.80 to 0.93, between domains). The instrument scores underwent a normalization process, revealing three distinct vulnerability strata: low (0 to 4), moderate (5 to 6), and high (7 to 14).

**Funding:** This work was supported by the Program of Support for the Institutional Development of the Unified Health System (Law No. 12,101, of November 27, 2009), through Ordinance No. 3,362, of December 8, 2017 –Technical Opinion No. 2 /2021 - CGGAP/DESF/SAPS/MS (0019478128) and dispatch SAPS/GAB/SAPS/MS (0019480381). The funders had no role in study design, data collection and analysis, decision to publish, or preparation of the manuscript.

**Competing interests:** The authors have declared that no competing interests exist.

## Conclusion

The scale exhibited satisfactory validity evidence, demonstrating consistency, reliability, and robustness. It resulted in a concise instrument that effectively measures and distinguishes levels of family vulnerability within the primary care setting in Brazil.

## Introduction

The term vulnerability is at the core of the debate in different knowledge fields; etymologically it comes from Latin "vulnerare" (harm, impair) and "bile" (susceptible) [1]. However, the understanding about vulnerability shows variations when different dimensions linked to this topic are considered [2]. Vulnerability, in the Bioethics field, refers to being in danger or at risk due to individual weakness, which is a feature inherent to humans [3]. As for the healthcare field, this term has a broader meaning. It is linked to the recognition that humans are vulnerable to harm or risks within the health/illness continuum as a result of social disadvantages [4].

Although the literature presents different definitions for vulnerability and assesses individual predictors, the concept of family vulnerability is measured in different ways [4]. It is crucial to consider this phenomenon from multiple perspectives that are interconnected with the health needs of individuals within a given family. Factors such as the health status of family members, the community environment, and the social context should all be taken into consideration when examining vulnerability within families [5,6].

Thus, health services providers and managers must consider family vulnerability in order to structure healthcare practices, particularly from a population perspective [7]. In line with this, adopting a territorial perspective that considers vulnerability strata can support the identification of health needs in various population groups [2]. Besides, family vulnerability stratification is essential to consider when planning the services to offer in a given territory, since it may help achieving equity and qualified population-based care management. Therefore, in order to align the activities of healthcare teams and the order of care provision within the Health Care Network, it is essential to further explore the elements that contribute to family vulnerability. This deeper understanding will enable a more comprehensive response to the demands arising from family vulnerability.

There is currently a scarcity of tools and initiatives that specifically address the incorporation of aspects related to family vulnerability as integral components within labor process organizations. Some global experiences are linked to this phenomenon: vulnerability-measuring background is in the United Nations Program for Development (UNPD). In 2004, the Disaster Risk Index (DRI) was created to measure and compare countries within a process based on physical, social, economic, and environmental factors [8]. Subsequently, in 2006, the Autonomous University of Madrid (Spain) conducted an estimation of the vulnerability levels among citizens who were assumed to be at risk of lacking social protection. This estimation was based on countries that are part of the Organization for Economic Cooperation and Development (OECD) [9]. In Latin America, one research aimed at measuring family vulnerability rates in a Colombian municipality based on a sample of families from all socioeconomic strata living in urban and rural zones [10].

Other initiatives have aimed at developing tools to be used by primary healthcare (PHC) teams to measure family vulnerability and, consequently, to contribute to plan healthcare provision in the Brazilian territory [11–15]. However, the tools so far developed and used for such a purpose still need robust evidence of their validity, as they present limitations to be used in a

country with continental dimensions, and multiple socioeconomic and cultural realities like Brazil.

Therefore, it is crucial to contemplate the concept of family vulnerability, with a particular emphasis on diverse populations. From this perspective, there is a need to develop an instrument that can be utilized in a standardized manner at a national level. Therefore, the present study aims to seek validity evidence about the content and internal structure of the Family Vulnerability Scale for Brazil (EVFAM-BR, in Portuguese).

## Materials and methods

The present study followed a psychometric design to gather evidence on both content validity (stage 1) and internal structure (stage 2), based on current recommendations by the Educational Research Association (AERA), the American Psychological Association (APA) and the National Council on Measurement in Education (NCME) [16]. The study was approved by the Ethics Research Committee of *Hospital Israelita Albert Einstein* on October 22, 2019 (n° 3.674.106, CAAE 12395919.0.0000.0071).

### Stage 1: Content validity evidence

PHC professionals (such as physicians, nurses, nursing assistants, community health agents, oral health team and multi-professional team, as psychologists, social workers, speech therapists, physiotherapists, nutritionists, and pharmacists) from all geographic regions in Brazil were invited to join the first qualitative exploratory stage of the study to define the concept of "family vulnerability" and to identify factors likely associated with it. The insights gathered from this stage were used to inform the development of items for the instrument. This effort was done to identify different views of family vulnerability in different geographic regions countrywide.

The invitation was conducted using a snowball sampling method [17], through an online questionnaire on the RedCap® electronic tool [18,19] sent via WhatsApp and email. After reading the Informed Consent Form (ICF) and providing consent to participate in the study, participants were given access to a semi-structured questionnaire.

The questionnaire comprised (i) respondents' socioeconomic, demographic and health profile identification, (ii) open questions about the concept of vulnerability and scale applicability, and (iii) multiple-choice questions about the relevance of measuring family vulnerability. The multiple-choice questions were distributed into 46 items elaborated from individual and domestic registration forms used at the Brazilian public healthcare system through e-SUS PHC software [20,21]. Frequencies of responses for items expressed in multi-choice questions and the group of aspects identified in open questions were taken into consideration according to the Content Analysis Technique, in order to elaborate the first version of identified items [22]–they were developed as questions to allow dichotomous answers ("0 –no" and "1 –yes").

Items developed from the previous stage were subjected to an extensive panel of multi-regional and multi-disciplinary judges. The panel encompassed health professionals, scholars and psychometrists who were invited to join the study through the snowball method and like previously, an online RedCap® questionnaire with ICF was used.

Registered participants accepted to participate and signed an ICF, but did not access the questionnaires related to the family vulnerability concept in the first qualitative exploratory stage or the questionnaire used to evaluate the items in the initial version of the instrument during the panel of judges were excluded from the study.

The large number of judges was explained by the need to adjust and apply the instrument at national level. The judges evaluated the items in the first version of the instrument based on

criteria of relevance and clarity, and were also asked whether there was a need to change the wording of any item.

Then, a choice was made to apply the Content Validity Ratio (CVR) [23] as the validity index to select the items. This choice was made because CVR is calculated based on the number of judges in the panel [24,25] and therefore more judges could be included. CVR was initially applied to assess the relevance of a given item in order to check whether it effectively measures the latent variable: family vulnerability. CVR was represented as CVR-1 (the Item's CVR) and CVR-E (Scale CVR)–this last one corresponds to mean recorded for the CVR criteria. It is important to highlight that a modified CVR version with two points, namely "no" and "yes", was adopted. The original version had three points and did not have an effective practical effect, since, for CVR calculation purposes, it turned dichotomous. This procedure has been adopted in other studies [26,27].

Oftentimes, the mean recorded for the assessed questionnaires was adopted; therefore, a hierarchic flow was adopted, since other indicators, such as item's clarity, were only analyzed after judges showed its relevance to testify that the item actually assessed the instrument's latent variable. The application of requirements in the same stage tends to inflate mean CVR and to launch items that do not measure the latent variable in the next stage. Accordingly, as pointed out by DeVellis [28], a given item can be relevant, but its words might be problematic. Thus, the second stage refers to items' clarity (whether the item is well written in terms of its semantics). The third stage assessed the need of changing the items' writing. The mean recorded for CVR was only applied to items that adhered to the phenomenon.

## Stage 2: Evidence about internal structure validity

The version subjected to content evidence was applied to users of PHC services to find evidence of internal structure validity.

All data collectors were previously trained and received guidance about the ICF application, as well as the research aim, methodology and questions. Study presentation and data collection flow were also previously carried out with teams from the participating PHC services.

PHC services selection was based on municipalities with the largest population in which the methodology of Health Care Planning was being implemented [29]. This was done by including at least one PHC service from each geographic region in the country. Thus, data collection was carried out in 11 PHC services: one in Northern Brazil (Roraima State), one in the North-eastern region (Pernambuco State) and two in Midwestern Brazil (Mato Grosso State), five in the Southeastern region (São Paulo and Minas Gerais states) and two in Southern Brazil (Paraná State).

Due to the Covid-19 pandemic, during the first data collection period (June to November 2020), interviews were conducted via telephone contact with healthcare users in São Paulo. This was done with the consent of the service management, who provided the identifying information of individuals within the territory. As for the second data collection time (from May to August 2022), users who had attended the participating PHC services at data-collection day were asked to join the study.

PHC users over 18-years-old were invited to participate. The Informed Consent Form approved by Hospital Israelita Albert Einstein in Ethics Research Committee was applied by the data collector. Verbal acceptance or refusal was registered using the RedCap® software on a tablet device and the participant received a printed copy of the ICF. Next, the structured questionnaire about the family vulnerability scale was applied and registered offline using the RedCap® installed on a tablet device. At the end of each day, information registered on each tablet was uploaded to the RedCap® server.

PHC users who accepted to participate according to the ICF but had missing information regarding the family vulnerability scale were excluded.

## Statistical analysis

**Exploratory factor analysis.** The first stage of the analysis aimed at assessing whether the collected data were prone to factorial through Measure of Sampling Adequacy (MSA). Bartlett sphericity, which is determinant of the matrix and Kaiser-Meyer-Olkin (KMO), were assessed at this stage. Besides assessing the dataset items, an individual analysis was also assessed, as recommended by Lorenzo-Seva and Ferrando [30]. The inadequacy of items to be factored can affect model solution. Missing data were treated with the multiple imputation technique [31].

Dimensionality testing was carried out through Parallel Analysis, based on Optimal implementation of Parallel Analysis (PA) and Minimum rank factor analysis to minimize the common variance of residues [32]. PA was implemented through permutation with 500 random matrices. Dimensionality in exploratory factorial analysis (unrestricted model) was tested through Parallel Analysis, which has been considered more robust and accurate to test it [33–37].

Tetrachoric matrix estimates were carried out through Bayes Modal Estimation [38], with Smoothing Ridge [39]. The use of tetrachoric/polychoric correlations tends to increase the model's accuracy in comparison to Pearson's correlation [40,41].

Factors' extraction was performed through the RULS technique (Robust Unweighted Least Squares), which reduces the residues in matrices that are more robust in terms of abnormal data [42]. Promin oblique rotation would be used in case the instrument emerged as multidimensional [43].

UNICO (Unidimensional Congruence > 0.95), ECV (Explained Common Variance > 0.80 –Quinn, 2014) and MIREAL (Mean of Item Residual Absolute Loadings < 0.30) were adopted as unidimensionality assessment indicator [44].

**Quality parameters of the instrument.** The tool variance must be close to 60% [45]. Initial factorial load of 0.30 is recommended when the sample comprises less than 300 individuals [45]; communities must present values higher than 0.40 [46].

The maintenance or removal of the item in the model will depend on the magnitude of the factorial loads, the communality, the absence of cross-loading, Heywood cases, and the interpretability of the factors.

The unique directional correlation (Eta) through Pratt's Measure was adopted to increase the accuracy of decision-making about the retention or removal of a given item [47,48].

**Reliability.** Reliability was measured through four indicators: Cronbach's alpha [49], Greatest Lower Bound–glb [50], Omega [51]—all three by means of Bayesian estimates—and ORION (Overall Reliability of Fully-Informative prior Oblique N-EAP scores) [52].

Cross-validation was applied to increase the model's reliability and replicability; the Houdolt technique was also applied [53]. This technique divides the dataset into a training sample —that can range from 10%, 30% to 50%—and into a dataset known as test dataset [53]. The dataset in the present study was split in half by randomly choosing the items. The Solomon technique [54] was adopted, so that dataset division could be random and respect factorability's equivalence. The datasets were labeled as follows: Full Sample (FS; n = 1,255); Training Sample (TrS n = 627) and Test Sample (TsS; n = 628). According to Brown [55], cross-validation can be carried out either through EFA or Confirmatory Factor Analysis (CFA). FS analysis will only take place if the model found in TrS and TsS can be replicated. This procedure was already adopted in previous studies [56,57], and it follows current recommendations [58].

**Descriptive study and standardization.** An exploratory descriptive study of general scores recorded for the Family Vulnerability Scale was carried out after a solution for the

internal structure was found. Results recorded for the items and for total score were represented by the frequency of answer, median (Md), interquartile interval (IIQ), amplitude (amp), minimum (min) and maximum (max) value.

In the first stage, standardization was performed by identifying score cutoffs based on the distribution of participants. Despite this process, the fact that participants' distribution is recurrent in standardization studies can lead to distortions, because the score is not directly analyzed, but considered according to participants' positions according to cutoff points. To achieve greater precision regarding the proposed cutoffs in the ranges and to assess the predictive capacity of individual classification, a discriminant analysis was used for each of the ranges and the scores of the Family Vulnerability Scale.

Discriminant analysis of each one of the limits and Family Vulnerability Scale scores were used to improve the accuracy of proposed cutoffs (within the limit) and to assess the predictive ability to classify the individuals. The discriminant analysis aims to allow a better understanding of group differences and to predict the probability of an entity (individual or object) to perceive a specific class or group, based on several independent variables of the metrics [59]. Boedeker and Kearns [60] identified a better performance using the discriminating analysis, in comparison to many other techniques applied for the same purpose. Besides, it allows determining the independent variables that have the most significant impact on the differences observed in the mean score profiles in two or more groups [59]. Tabachnick and Fidell [61] added to this information by stating that the aim of the discriminating analysis is to predict the group's participation based on a set of predictors. Accordingly, it is possible to confirm whether the groups formed from the distribution process have properly classified individuals within the established limits.

Data were analyzed in statistic software Factor 12.01.01, SPSS v.23 and JASP 16.04.

## Results

### Content validity evidence

A total of 123 professionals from five Brazilian regions joined the first stage of the research to define the concept of "family vulnerability": 48.8% of them came from Northeastern Brazil; 21.1% from Southeastern Brazil; 17.9% from Southern Brazil, 8.9% from Northern Brazil and 3.3% from Midwestern Brazil. Most professionals were female (82.9%), approximately 40% of them had at least 10-year experience in PHC and 48.8% reported having a specialization degree as higher education level. Most participation in this stage were nurses (48.3%) and community health agents (10.7%). Based on the responses of the factors that could potentially be associated with the concept of family vulnerability, the first version of the instrument was developed, containing 92 items.

A panel of multi-regional and multi-disciplinary judges was set for the second stage; it aimed at identifying content validity evidence. This panel had 73 judges: 61.7% from Southeastern Brazil, 15.1% from Southern Brazil, 9.6% Northeastern Brazil, 6.8% from Northern Brazil and 6.8% from Midwestern Brazil. Most of them were female (79.5%), and had at least 10-year experience in PHC (57.5%) and a specialization degree as higher education level (51.4%). Most in the panel were nurses (50.7%) and physicians (16.55). CVR was applied to judges' answers. CVR critical value was established at CVR > 0.12, which was defined based on the participation of 73 judges.

CVR calculation led to rule out 54 items resulting in a scale with 38 items that are related to socioeconomic and demographic aspects, access to healthcare services, health condition, and lifestyle. To make the scale even clearer, 12 of the 38 items were rewritten based on recommendations from the panel of judges. Only items 2 and 15 (Table 1) recorded a CVR lower than

**Table 1. Items that remained in the scale after applying the Content Validity Ratio (CVR).**

| Item | Content Validity Ratio (CVR) | | | |
|---|---|---|---|---|
| | Relevance | Clarity | Need of change[a] | New text of the item |
| 1. Is there a lack of basic sanitation in the neighborhood you live in? | 0.12 | 0.12 | 0.78 | Is there open sewer in your neighborhood? |
| 2. Do you drink untreated water in your house? | 0.12 | 0.07 | 0.86 | Does the water in your house lack treatment? |
| 3. Is your household at risk of flood? | 0.18 | 0.40 | 1.00 | - |
| 4. Do you live close to drug dealing areas? | 0.12 | 0.40 | 0.95 | - |
| 5. Does anyone in the household coexist with violent individuals? | 0.21 | 0.32 | 0.97 | - |
| 6. Has anyone in your household been the victim of violence? | 0.26 | 0.34 | 1.00 | - |
| 7. Is there any violence happening in your home? | 0.23 | 0.32 | 1.00 | - |
| 8. Is any family member in an incarceration situation? | 0.12 | 0.26 | 0.86 | Is anyone in your family in jail? |
| 9. Is anyone in your household having financial difficulties? | 0.15 | 0.12 | 0.97 | - |
| 10. Is money short to meet household needs? | 0.12 | 0.21 | 0.97 | - |
| 11. Does anyone in your household is a *Bolsa Família* beneficiary? | 0.12 | 0.32 | 1.00 | - |
| 12. Is anyone in your household a BPC/LOAS (Continuous Cash Benefit or Social Assistance) beneficiary? | 0.15 | 0.21 | 0.86 | Does anyone in your household receives health benefit (BPC/LOAS)? |
| 13. Has any health professional ever mentioned that someone in your household is obese? | 0.12 | 0.15 | 0.86 | - |
| 14. Has any health professional ever mentioned that someone in your household suffers with malnutrition? | 0.15 | 0.12 | 0.89 | - |
| 15. Is there any difficulties securing access to a variety of foods at home? | 0.15 | 0.10 | 0.92 | It is hard to secure access to different food types? |
| 16. Does anyone in your household starve? | 0.21 | 0.26 | 0.92 | - |
| 17. Does anyone in your household have drug addiction? | 0.23 | 0.26 | 0.89 | Does anyone in your household use illegal drugs? |
| 18. Is anyone in your household an alcohol abuser? | 0.23 | 0.26 | 0.92 | - |
| 19. Does anyone in your household use medication? | 0.15 | 0.37 | 0.95 | - |
| 20. Does anyone in your household use five or more medications daily? | 0.18 | 0.40 | 0.95 | Does anyone in your household use five types of medications, or more, daily? |
| 21. Does anyone in your household have a health condition that demands long-term caregiving? | 0.21 | 0.26 | 0.97 | Does anyone in your household have a health condition that requires continuous care? |
| 22. Is anyone in your household impaired to perform daily activities? | 0.18 | 0.23 | 0.86 | - |
| 23. Is anyone in your house helped by others to accomplish daily healthcare procedures? | 0.15 | 0.21 | 0.95 | Does anyone in your household need help accomplishing the daily healthcare procedures? |
| 24. Does anyone in your household present any disability? | 0.12 | 0.37 | 1.00 | - |
| 25. Does anyone in your household have any intellectual/cognitive disability? | 0.15 | 0.29 | 0.84 | - |
| 26. Does anyone in your household have mental health issues? | 0.21 | 0.34 | 1.00 | - |
| 27. Does anyone in your household have HIV/AIDS? | 0.15 | 0.32 | 0.97 | - |
| 28. Is anyone in your household bedbound? | 0.15 | 0.32 | 1.00 | - |
| 29. Does anyone in your household often go to urgency and emergency units? | 0.18 | 0.29 | 0.95 | - |
| 30. Does anyone in your household unaware of who the UBS/healthcare unit team in charge of your family is? | 0.15 | 0.32 | 0.95 | - |
| 31. Did anyone in your household have a child without wanting it? | 0.15 | 0.29 | 0.95 | Has anyone in your household had an unplanned child? |
| 32. Did anyone in your household have a child before turning 20 years old? | 0.12 | 0.37 | 0.97 | - |
| 33. Did anyone in your household have an absent mother in childhood? | 0.18 | 0.34 | 1.00 | - |
| 34. Did anyone in your household have an absent father in t childhood? | 0.12 | 0.32 | 1.00 | - |
| 35. Has anyone in your household been abandoned by the family? | 0.21 | 0.18 | 0.95 | - |
| 36. Are children in your household out of school? | 0.12 | 0.32 | 0.92 | Are there children in your household out of school? |

*(Continued)*

**Table 1.** (Continued)

| Item | Content Validity Ratio (CVR) | | | |
|------|------------|---------|----------------------|----------------------|
| | Relevance | Clarity | Need of change[a] | New text of the item |
| 37. Are there adolescents in your household out of school? | 0.12 | 0.37 | 0.97 | Do you have any adolescent in your household out of school? |
| 38. Are there under 14-year-old individuals in your household who have a job? | 0.21 | 0.23 | 0.95 | - |
| **Mean CVR** | **0.16** | **0.27** | **0.94** | - |

[a] Accordingly, CVR values higher than the critical value indicate no need of changing the writings, although some items under this condition were rewritten in order to improve semantics' adequacy.

the critical value; therefore, their text was revised. The other 10 items did not suffer any changes; writing adjustments were made in the original text just to meet CVR's critical value.

The version containing evidence of content validity (38 items) was considered in the stage to demonstrate internal structure validity.

## Evidence about internal structure validity

In total, 1,584 users who attended the 11 PHC services during data collection were invited to join this research stage. However, only 1,505 (95%) of them accepted the invitation and signed the informed consent form. Only 1,255 completed the interview for the application of the scale version presenting content validity evidence. This sample represents the study final sample and this version presents content validity evidence. Table 2 introduces participants descriptions of this study stage.

**Factorability.** The evaluation of sample adequacy measures is the first step in the factorability analysis; it aims at assessing dataset factorability and whether factorial analyses are applicable. General database data have shown good factorability: Fs recorded KMO (0.75), Bartlett Sphericity = 6,084.6 (df = 91; P < 0.0001) and determinant of the matrix = 0.00001. As for TrS: KMO (0.74), Bartlett Sphericity = 6,153.7 (df = 91; P < 0.0001) and determinant of the matrix = 0.00001; TsS: KMO (0.71), Bartlett Sphericity = 2,617.3 (df = 91; P < 0.0001) and determinant of the matrix = 0.0002. Although the general indices presented good indicators, 4 of the 38 initial items have shown factorability issues in three datasets (27—Does anyone in your household have HIV/AIDS?; 36—Are there children in your household out of school?; 37 —Do you have any adolescent in your household out of school?; and 38—Are there under 14-year-old individuals in your household who have a job?). They were excluded from the analyses based on recommendations by Lorenzo-Seva and Ferrando [30].

**Dimensionability.** The first analyses were carried out in TrS. Dimensionality analyzed through parallel analysis pointed towards a 4-dimension model. Closeness of dimensionality values kept the indication for multi-dimensional model: Single = 0.82; ECV = 0.65 and MIREAL = 0.37. Thirteen (13) of the 34 items forming the initial analysis did not present substantial factorial load in the model. Accordingly, the process to remove items to adjust the model followed two principles: quantitative (statistical adjustment) and qualitative (interpretability) adjustment. The choice for removing an item was carried out by taking into consideration the set of primary indicators: factorial load, communality, Eta of Pratt's Importance Measure, existence of cross-loading, Heywood case and model adjustment indices. The items were removed from the scale until the two principles were congruent, resulting in a model for 4-dimension TrS with 14 items showing appropriate statistical fit and interpretability.

**Table 2. Participants demographics.**

| Variables (n = 1255) | Categories | N (%) |
|---|---|---|
| Age in years[a] | - | 43.3 (15.5%) |
| Sex (n = 756) | Female | 551 (43.9%) |
| | Male | 205 (16.3%) |
| Race/skin color (n = 1217) | White | 386 (30.8%) |
| | Brown | 640 (51.0%) |
| | Black | 150 (12.0%) |
| | Yellow | 23 (1.8%) |
| | Indigenous | 18 (1.4%) |
| Schooling (in years) (n = 1237) | 0 to 4 years | 160 (12.7%) |
| | 5 to 8 years | 225 (17.9%) |
| | 9 to 11 years | 247 (19.7%) |
| | 12 to 15 years | 480 (38.2%) |
| | Over 16 years | 125 (10.0%) |
| Job (n = 1237) | Unemployed or does not have a job | 447 (35.6%) |
| | Employer | 1 (0.1%) |
| | Self-employed without social security assistance | 111 (8.8%) |
| | Self-employed with social security assistance | 47 (3.7%) |
| | Wage earner | 413 (32.9%) |
| | Retired/pensioner | 181 (14.4%) |
| | Others | 37 (2.9%) |
| Income (in minimum wage)[b] (n = 726) | No income | 58 (4.6%) |
| | Up to 1 minimum wage | 327 (26.1%) |
| | > 1 and lower than 2 minimum wages | 179 (14.3%) |
| | > = 2 and lower than 4 minimum wages | 115 (9.2%) |
| | > = 4 lower than 10 minimum wages | 31 (2.5%) |
| | > = 10 minimum wages | 4 (0.3%) |
| | Does not know | 12 (1.0%) |
| Number of children (n = 759) | 1 child | 140 (11.2%) |
| | 2 children | 198 (15.8%) |
| | 3 children | 129 (10.3%) |
| | More than 3 children | 129 (10.3%) |
| | Expecting the first child | 19 (1.5%) |
| | Does not have children | 144 (11.5%) |
| Number of households per room in the house (n = 1223) | <1 | 406 (32.4%) |
| | 1 | 697 (55.5%) |
| | >1 | 120 (9.6%) |
| Private healthcare insurance (n = 1231) | Yes | 180 (14.3%) |
| | No | 1,051 (83.7%) |
| Systemic High Blood Pressure (n = 1240) | Yes | 212 (16.9%) |
| | No | 1,028 (81.9%) |
| Diabetes Mellitus (n = 1238) | Yes | 96 (7.6%) |
| | No | 1,142 (91%) |
| Cancer (current) (n = 1237) | Yes | 6 (0.5%) |
| | No | 1,231 (98.1%) |
| Heart disease (n = 1240) | Yes | 72 (5.7%) |
| | No | 1,168 (93.1%) |

(*Continued*)

**Table 2.** (Continued)

| Variables (n = 1255) | Categories | N (%) |
|---|---|---|
| Intellectual/cognitive impairment (n = 812) | Yes | 15 (1.2%) |
| | No | 797 (63.5%) |
| Tuberculosis (n = 1240) | Yes | 3 (0.2%) |
| | No | 1,237 (98.6%) |
| Leprosy (n = 1239) | Yes | 1 (0.1%) |
| | No | 1,238 (98.6%) |
| Kidney issues (n = 1237) | Yes | 35 (2.8%) |
| | No | 1,202 (95.8%) |
| Breathing issues (n = 1232) | Yes | 96 (7.6%) |
| | No | 1,136 (90.5%) |
| Mental health diagnosis (n = 1238) | Yes | 30 (2.4%) |
| | No | 1,208 (96.3%) |

[a] continuous numerical variable described as mean and standard deviation.
[b] the minimum wage corresponds to R$1,212.00 (in Brazilian Real, in 2022).

The parallel analysis kept on pointing out a 4-dimension solution and explained variance of 78.66%. This model was replicated in TsS and FS. Both datasets confirmed the 4-dimension model; furthermore, the closeness of dimensionality values reinforced the multi-dimensional model (Table 3) for the three datasets.

TrS primary data (Table 4) presents factorial data ranging from 0.597 to 0.975, communality ranging from 0.449 to 0.967, and Eta ranging from 0.640 to 0.941. The model recorded explained variable of 76.18%. Items 1 and 3 comprised the dimension **Income,** items from 4 to 8 comprised the dimension **Healthcare**, dimension **Family** was in items 9 to 11, and the single dimension **Violence** was observed in items 12 to 14. Accordingly, the model points towards

**Table 3. I-Unico, I-ECV, and I-Real item values.**

| Item | I-UNICO | | | I-ECV | | | I-REAL | | |
|---|---|---|---|---|---|---|---|---|---|
| | TrS | TsS | FS | TrS | TsS | FS | TrS | TsS | FS |
| 1. Is anyone in your household having financial difficulties? | 1.000 | 0.958 | 1.000 | 0.991 | 0.770 | 0.984 | 0.063 | 0.352 | 0.084 |
| 2. Is money short to meet household needs? | 1.000 | 0.931 | 1.000 | 0.996 | 0.719 | 0.990 | 0.038 | 0.389 | 0.061 |
| 3. Is it hard to secure access to different food types? | 1.000 | 0.988 | 1.000 | 0.976 | 0.865 | 0.992 | 0.094 | 0.232 | 0.053 |
| 4. Does anyone in your household use medication? | 0.957 | 0.983 | 0.970 | 0.766 | 0.844 | 0.799 | 0.328 | 0.275 | 0.320 |
| 5. Does anyone in your household use five types of medications, or more, daily? | 0.662 | 0.988 | 0.926 | 0.469 | 0.863 | 0.711 | 0.491 | 0.261 | 0.361 |
| 6. Does anyone in your household have a health condition that requires continuous care? | 0.927 | 0.968 | 0.963 | 0.711 | 0.795 | 0.782 | 0.449 | 0.383 | 0.374 |
| 7. Is anyone in your household impaired to perform daily activities? | 0.850 | 0.966 | 0.941 | 0.617 | 0.789 | 0.735 | 0.560 | 0.400 | 0.439 |
| 8. Does anyone in your household need help accomplishing daily healthcare procedures? | 0.742 | 0.961 | 0.887 | 0.525 | 0.777 | 0.657 | 0.538 | 0.378 | 0.478 |
| 9. Did anyone in your household have an absent mother in childhood? | 0.291 | 0.347 | 0.387 | 0.233 | 0.270 | 0.295 | 0.600 | 0.493 | 0.516 |
| 10. Did anyone in your household have an absent father in childhood? | 0.495 | 0.240 | 0.511 | 0.363 | 0.198 | 0.373 | 0.465 | 0.535 | 0.389 |
| 11. Has anyone in your household been abandoned by the family? | 0.927 | 0.583 | 0.912 | 0.711 | 0.418 | 0.690 | 0.329 | 0.439 | 0.297 |
| 12. Does anyone in the household coexist with violent individuals? | 0.868 | 0.936 | 0.488 | 0.636 | 0.727 | 0.358 | 0.481 | 0.115 | 0.616 |
| 13. Has anyone in your household been the victim of violence? | 0.861 | 0.914 | 0.731 | 0.628 | 0.693 | 0.517 | 0.450 | 0.295 | 0.564 |
| 14. Is there any violence happening in your home? | 0.914 | 0.963 | 0.349 | 0.692 | 0.782 | 0.272 | 0.381 | 0.021 | 0.503 |

I-UNICO, Unidimensional Congruence; I-ECV, Explained Common Variance; I-REAL, Residual Absolute Loadings; TrS, Training Sample; TsS, Test Sample; FS, Full Sample.

**Table 4. Training dataset: Factorial loads, communality, and Eta.**

| Item | Factorial Load | | | | h² | Pratt's Measure—(Eta) | | | |
|---|---|---|---|---|---|---|---|---|---|
| | Income | Healthcare | Family | Violence | | Income | Healthcare | Family | Violence |
| 1. Is anyone in your household having financial difficulties? | **0.893** | 0.027 | -0.012 | 0.059 | 0.849 | **0.904** | 0.098 | 0.000 | 0.146 |
| 2. Is money short to meet household needs? | **0.895** | -0.013 | 0.013 | -0.067 | 0.762 | **0.872** | 0.000 | 0.049 | 0.000 |
| 3. Is it hard to secure access to different food types? | **0.597** | 0.091 | 0.102 | 0.110 | 0.516 | **0.642** | 0.180 | 0.174 | 0.203 |
| 4. Does anyone in your household use medication? | -0.100 | **0.693** | 0.101 | 0.067 | 0.500 | 0.000 | **0.680** | 0.146 | 0.130 |
| Does anyone in your household use five types of medications, or more, daily? | -0.080 | **0.711** | 0.031 | -0.118 | 0.449 | 0.000 | **0.668** | 0.051 | 0.000 |
| 6. Does anyone in your household have a health condition that requires continuous care? | -0.032 | **0.805** | 0.020 | 0.069 | 0.671 | 0.000 | **0.805** | 0.059 | 0.139 |
| 7. Is anyone in your household impaired to perform daily activities? | 0.094 | **0.862** | -0.131 | 0.029 | 0.798 | 0.189 | **0.869** | 0.000 | 0.084 |
| 8. Does anyone in your household need help accomplishing daily healthcare procedures? | 0.062 | **0.737** | -0.053 | -0.090 | 0.540 | 0.132 | **0.723** | 0.000 | 0.000 |
| 9. Did anyone in your household have an absent mother in childhood? | 0.012 | -0.140 | **0.798** | 0.013 | 0.630 | 0.042 | 0.000 | **0.790** | 0.061 |
| 10. Did anyone in your household have an absent father in childhood? | 0.056 | -0.033 | **0.741** | -0.097 | 0.514 | 0.099 | 0.000 | **0.710** | 0.000 |
| 11. Has anyone in your household been abandoned by the family? | 0.018 | 0.148 | **0.630** | 0.073 | 0.504 | 0.065 | 0.203 | **0.658** | 0.163 |
| 12. Does anyone in the household coexist with violent individuals? | 0.192 | -0.130 | -0.108 | **0.975** | 0.967 | 0.284 | 0.000 | 0.000 | **0.941** |
| 13. Has anyone in your household been the victim of violence? | -0.118 | 0.086 | 0.290 | **0.605** | 0.574 | 0.000 | 0.146 | 0.379 | **0.640** |
| 14. Is there any violence happening in your home? | -0.124 | 0.055 | -0.091 | **0.937** | 0.787 | 0.000 | 0.117 | 0.000 | **0.880** |

good factorial and interpretable (quantitatively) solution, with content alignment in coherent and interpretable (quantitatively) items. The final version of the scale represented reduction by approximately 63% in the 38 items assessed through judges' panel in the first stage. This value is close to that presented by DeVellis [28], according to whom the researcher must project items lost by 50% throughout the process.

Based on the four dimensions forming the scale, it is possible to determine the key role played by social determinants within the health/illness process. Dimensions embody items related to income, social and family cohesion, and to life and housing conditions associated with psychosocial and behavioral aspects [62]. Thus, they extrapolate the biological view of health, which is overall acknowledged in a reductionist way, focused at the medical practices [62–65]. Nevertheless, the present instrument emerges as a multi-disciplinary work-tool available for PHC teams that have the potential to promote social justice by taking into consideration social inequities when planning healthcare services.

Factorial loads recorded for the TsS dataset (Table 5) ranged from 0.643 to 0.976, communality ranged from 0.482 to 0.910 and Eta ranged from 0.658 to 0.951. This model presented explained variance of 76.18%. Once again, the observed model was equal to the training dataset model; consequently, it could be interpreted quantitatively and qualitatively.

In the FS dataset (Table 6), these results ranged from 0.308 to 0.785 for factorial loads, from 0.212 to 0.967 for communality, and Eta values from 0.640 to 0.941. The model using the entire sample achieved an explained variance of 79.02%. An essential aspect is the stability in the number of dimensions and the interpretability of the model, which reinforces the importance of conducting cross-validation.

The reliability indices across analysis datasets ranged from 0.69 to 0.71 for Cronbach's alpha, reaching 0.70 in all three datasets for Omega. Additionally, it ranged from 0.83 to 0.84 for glb and from 0.80 to 0.96 for ORION, indicating strong consistency between dimensions and datasets. The quality of the factorial solution indices also demonstrated adequate levels, further reinforcing the stability of the model (Table 7). Consequently, the combination of

**Table 5. Test dataset: Factor loading, communality and Eta.**

| Item | Factorial Load | | | | h² | Pratt's Measure—(Eta) | | | |
|---|---|---|---|---|---|---|---|---|---|
| | Income | Healthcare | Family | Violence | | Income | Healthcare | Family | Violence |
| 1. Is anyone in your household having financial difficulties? | **0.801** | 0.032 | 0.046 | 0.056 | 0.702 | **0.818** | 0.108 | 0.128 | 0.075 |
| 2. Is money short to meet household needs? | **0.976** | -0.072 | 0.013 | -0.008 | 0.910 | **0.951** | 0.000 | 0.068 | 0.000 |
| 3. Is it hard to secure access to different food types? | **0.805** | 0.063 | -0.084 | -0.006 | 0.647 | **0.790** | 0.151 | 0.000 | 0.000 |
| 4. Does anyone in your household use medication? | 0.059 | **0.671** | -0.009 | -0.006 | 0.482 | 0.138 | **0.681** | 0.000 | 0.000 |
| Does anyone in your household use five types of medications, or more, daily? | 0.141 | **0.639** | -0.078 | 0.066 | 0.493 | 0.227 | **0.658** | 0.000 | 0.094 |
| 6. Does anyone in your household have a health condition that requires continuous care? | -0.047 | **0.882** | 0.023 | -0.012 | 0.753 | 0.000 | **0.865** | 0.063 | 0.000 |
| 7. Is anyone in your household impaired to perform daily activities? | 0.001 | **0.881** | -0.030 | 0.011 | 0.770 | 0.015 | **0.876** | 0.000 | 0.035 |
| 8. Does anyone in your household need help accomplishing daily healthcare procedures? | -0.080 | **0.856** | 0.037 | -0.033 | 0.690 | 0.000 | **0.827** | 0.078 | 0.000 |
| 9. Did anyone in your household have an absent mother in childhood? | -0.024 | -0.030 | **0.727** | 0.036 | 0.514 | 0.000 | 0.000 | **0.715** | 0.059 |
| 10. Did anyone in your household have an absent father in childhood? | -0.015 | -0.090 | **0.752** | 0.037 | 0.546 | 0.000 | 0.000 | **0.736** | 0.059 |
| 11. Has anyone in your household been abandoned by the family? | -0.035 | 0.077 | **0.757** | -0.062 | 0.575 | 0.000 | 0.123 | **0.748** | 0.000 |
| 12. Does anyone in the household coexist with violent individuals? | 0.067 | -0.016 | -0.037 | **0.918** | 0.843 | 0.076 | 0.000 | 0.000 | **0.915** |
| 13. Has anyone in your household been the victim of violence? | 0.138 | 0.096 | 0.310 | **0.643** | 0.652 | 0.211 | 0.164 | 0.370 | **0.666** |
| 14. Is there any violence happening in your home? | -0.177 | -0.041 | -0.172 | **0.923** | 0.893 | 0.196 | 0.042 | 0.167 | **0.909** |

applied techniques and indices provided strong evidence of internal structure validity that is appropriate, consistent, robust, and interpretable.

**Normalization.** Once the internal structure of the instrument was established and validated, we proceeded with the descriptive study and normalization of scores to enable the interpretability of the instrument and proper classification of participants' scores. In this regard, Table 8 displays the response frequency for each questionnaire item. There is a clear prevalence of respondents selecting "No" for all items of the instrument. Some items had a

**Table 6. Full dataset: Factor loading, communality, and Eta.**

| Item | Factorial load | | | | h² | Pratt´s Measure—(Eta) | | | |
|---|---|---|---|---|---|---|---|---|---|
| | Income | Healthcare | Family | Violence | | Income | Healthcare | Family | Violence |
| 1. Is anyone in your household having financial difficulties? | **0.733** | 0.009 | 0.004 | 0.023 | 0.548 | **0.736** | 0.049 | 0.029 | 0.047 |
| 2. Is money short to meet household needs? | **0.785** | -0.045 | -0.015 | -0.020 | 0.585 | **0.765** | 0.000 | 0.000 | 0.000 |
| 3. Is it hard to secure access to different food types? | **0.555** | 0.059 | 0.019 | 0.027 | 0.346 | **0.569** | 0.124 | 0.060 | 0.050 |
| 4. Does anyone in your household use medication? | -0.001 | **0.523** | 0.056 | 0.016 | 0.287 | 0.000 | **0.528** | 0.087 | 0.035 |
| 5. Does anyone in your household use five types of medications, or more, daily? | 0.019 | **0.489** | -0.012 | -0.010 | 0.243 | 0.060 | **0.489** | 0.000 | 0.000 |
| 6. Does anyone in your household have a health condition that requires continuous care? | 0.007 | **0.639** | 0.032 | 0.013 | 0.421 | 0.041 | **0.644** | 0.065 | 0.032 |
| 7. Is anyone in your household impaired to perform daily activities? | 0.008 | **0.723** | -0.048 | 0.001 | 0.518 | 0.044 | **0.718** | 0.000 | 0.008 |
| 8. Does anyone in your household need help accomplishing daily healthcare procedures? | -0.036 | **0.624** | -0.017 | -0.024 | 0.371 | 0.000 | **0.609** | 0.000 | 0.000 |
| 9. Did anyone in your household have an absent mother in childhood? | -0.027 | -0.049 | **0.594** | 0.004 | 0.339 | 0.000 | 0.000 | **0.582** | 0.013 |
| 10. Did anyone in your household have an absent father in childhood? | 0.009 | -0.043 | **0.552** | -0.009 | 0.302 | 0.037 | 0.000 | **0.548** | 0.000 |
| 11. Has anyone in your household been abandoned by the family? | 0.004 | 0.060 | **0.471** | 0.001 | 0.236 | 0.026 | 0.089 | **0.476** | 0.007 |
| 12. Does anyone in the household coexist with violent individuals? | 0.025 | -0.034 | 0.011 | **0.743** | 0.554 | 0.047 | 0.000 | 0.029 | **0.742** |
| 13. Has anyone in your household been the victim of violence? | 0.028 | 0.084 | 0.273 | **0.308** | 0.212 | 0.069 | 0.119 | 0.296 | **0.325** |
| 14. Is there any violence happening in your home? | -0.047 | -0.005 | -0.018 | **0.597** | 0.351 | 0.000 | 0.000 | 0.000 | **0.593** |

**Table 7. Synthesis of the models.**

| Synthesis | Index | Technique | Training Sample (TrS) | Test Sample (TsS) | Full Sample (FS) |
|---|---|---|---|---|---|
| Exploratory | Adequacy of correlation matrix | Determinant of the matrix | < 0.000001 | < 0.000001 | 0.0002 |
| | | Bartlett | 6084.6 (df = 91) | 6153.7 (df = 91) | 8936.7 |
| | | KMO (Kaiser-Meyer-Olkin) | 0.75 | 0.74 | 0.71 |
| | Explained Variance (AP) | | 77.73% | 76.18% | 79.02% |
| | Polychoric Correlation ($r_p$ =) | | -0.08 to 0.80 | -0.21 to 0.81 | -0.04 to 0.82 |
| Reliability | Cronbach's Alpha | | 0.69 | 0.71 | 0.71 |
| | McDonald's Omega | | 0.70 | 0.70 | 0.70 |
| | Greatest Lower Bound–glb | | 0.83 | 0.84 | 0.83 |
| | ORION[a] | | 0.90; 0.96; 0.80; 0.91 | 0.81; 0.91; 0.93; 0.93 | 0.80; 0.93; 0.90; 0.92 |
| Unidimensional Assessment | Unidimensional Congruence (UNICO) | | 0.82 | 0.83 | 0.79 |
| | Explained Common Variance (ECV) | | 0.65 | 0.70 | 0.66 |
| | Mean of item residual absolute loading (MIREAL) | | 0.37 | 0.32 | 0.36 |
| Quality and Effectiveness | Factor Determinacy Index (FDI)[a] | | 0.94; 0.98; 0.89; 0.95 | 0.90; 0.95; 0.96; 0.96 | 0.89; 0.96; 0.95; 0.95 |
| | Sensitivity Ratio (SR)[a] | | 3.02; 5.37; 2.04; 3.20 | 2.09; 3.29; 3.77; 3.90 | 2.00; 3.74; 3.11; 3.37 |
| | Expected percentage of true differences (EPTD)[a] | | 92.6%; 96.6%; 88.9%; 93.0% | 89.1%; 93.3%; 94.3%; 94.6% | 88.7%; 94.2%; 92.8%; 93.5% |

[a] from dimension 1 to 4, respectively.

higher frequency of "Yes" responses, such as "Is money short to meet household needs?" with 40.34%, "Does anyone in the household use medication?" with 36.41%, "Does anyone in the household have a health condition that requires continuous care?" with 36.77%, and "Did anyone in the household have an absent father during childhood?" with 34.07%.

Table 9 presents the scores recorded for the dimensions and the general score of the Family Vulnerability Scale. All dimensions had all their amplitudes answered. Dimensions Income, Family and Violence recorded median = 0, Healthcare showed median = 1 and total score recorded median = 2. An interesting aspect of the total score lies on the fact that the amplitude

**Table 8. Item answer frequency.**

| Item | Answer frequency (N/%) | | |
|---|---|---|---|
| | No | Yes | Missing |
| 1. Is anyone in your household having financial difficulties? | 870 (69.10) | 377 (29.94) | 12 (0.95) |
| 2. Is money short to meet household needs? | 742 (58.93) | 508 (40.34) | 9 (0.71) |
| 3. Is it hard to secure access to different food types? | 928 (73.70) | 322 (25.57) | 9 (0.71) |
| 4. Does anyone in your household use medication? | 793 (62.98) | 461 (36.61) | 5 (0.39) |
| 5. Does anyone in your household use five types of medications, or more, daily? | 1,020 (81.01) | 231 (18.34) | 8 (0.63) |
| 6. Does anyone in your household have a health condition that requires continuous care? | 788 (62.58) | 463 (36.77) | 8 (0.63) |
| 7. Is anyone in your household impaired to perform daily activities? | 1,026 (81.43) | 232 (18.42) | 1 (0.07) |
| 8. Does anyone in your household need help accomplishing daily healthcare procedures? | 1,066 (84.67) | 190 (15.09) | 3 (0.23) |
| 9. Did anyone in your household have an absent mother in childhood? | 1,048 (83.24) | 207 (16.44) | 4 (0.31) |
| 10. Did anyone in your household have an absent father in childhood? | 826 (65.60) | 429 (34.07) | 4 (0.31) |
| 11. Has anyone in your household been abandoned by the family? | 1,129 (89.67) | 125 (9.92) | 5 (0.39) |
| 12. Does anyone in the household coexist with violent individuals? | 1,221 (96.98) | 34 (2.70) | 4 (0.31) |
| 13. Has anyone in your household been the victim of violence? | 1,088 (86.41) | 167 (13.26) | 4 (0.31) |
| 14. Is there any violence happening in your home? | 1,232 (97.85) | 25 (1.98) | 2 (0.15) |

**Table 9. Description of dimensions and scores recorded for the Family Vulnerability Scale.**

| Dimension / Score | Central Trend Measurements and Dispersion | | | | |
|---|---|---|---|---|---|
| | **Median** | **Minimum** | **Maximum** | **Amplitude** | **Interquartile** |
| Income dimension | 0.00 | 0.00 | 3 | 3 | 2.00 |
| Healthcare dimension | 1.00 | 0.00 | 5 | 5 | 2.00 |
| Family dimension | 0.00 | 0.00 | 3 | 3 | 1.00 |
| Violence Dimension | 0.00 | 0.00 | 3 | 3 | 0.00 |
| Total Score Total | 2.00 | 0.00 | 12 | 12 | 3.00 |

ranged from 0 to 14 and the maximum score recorded in the current sample reached 12. Medians in the minimum limit, and close to it, had previously indicated that the instrument could accurately differentiate individuals eventually facing family vulnerability situations.

Because these scores are closer to the bottom limit, they started to be impactful only when they move away from the median that, in this case, was close to the bottom limit and thus in the upper quartile. Thus, three initial classifications were suggested: model 1 had a cutoff in the median (low and high vulnerability), model 2 scores were separated up to percentile 75, from 76 to 89, and 90 and above; model 3 scores were separated up to percentile 70, from 70 to 89, or higher.

The discriminant analysis of each classification developed to assess whether it was possible to accurately identify participants within the limit was applied after the first cutoffs were made.

The first analysis adopted a binary classification (low and high). The discriminant analysis showed MBox = 446.58 p < 0.001. $\lambda_{wilks}$ = 0.36; $F_{(1, 1257)}$ = 1,268.77; p < 0.001; canonical correlation = 0.797; the model with two limits properly classified 85.7% of the cases. The discriminant analysis applied to model 2 was MBox = 49.64 p < 0.001. $\lambda_{wilks}$ = 0.18; $F_{(2, 1256)}$ = 2,094.25; p < 0.001; canonical correlation = 0.907. Model 2 properly classified 100% of cases. The analysis applied to model 3 presented MBox = 49.64 p < 0.001. $\lambda_{wilks}$ = 0.25; $F_{(2, 1256)}$ = 1,838.71; p < 0.001; canonical correlation = 0.863; it was possible to properly classify 89% of the cases. The recommended classification and scores interpretations are depicted in Table 10.

## Family Vulnerability Scale for Brazil–final version

The final version of the Family Vulnerability Scale for Brazil (EVFAM-BR) comprised 14 items (or questions) applied to each family in the PHC territory in Brazil, due to the action by community health agents. EVFAM-BR has four dimensions; each one has a score corresponding to the number of items in the dimension; at the end of its application, the score must range from zero (0) to fourteen (14). Healthcare is the dimension accounting for the largest number of items; consequently, it has the greatest scoring potential on the scale (n = 5).

Through the sum of scores for each dimension, the final score of EVFAM-BR presents three classification ranges of family vulnerability: Low (0 to 4), Moderate (5 to 6), and High (7 to 14). Table 11 (in English) and Table 12 (in Portuguese) summarize the EVFAM-BR, with their respective dimensions, items, and scores.

**Table 10. Limits, classification, and interpretation of Family Vulnerability Scale scores.**

| Classification results | Percentile | Name | Score |
|---|---|---|---|
| Limit | Up to 75 | low | 0 to 4 |
| | 76 to 89 | Moderate | 5 to 6 |
| | Higher than 90 | High | Higher than 7 |

**Table 11. Family Vulnerability Scale for Brazil (EVFAM-BR).**

| Dimension | Item | Item score | Dimension score |
|---|---|---|---|
| Income | 1. Is anyone in your household having financial difficulties? | 1 | 3 |
| | 2. Is money short to meet household needs? | 1 | |
| | 3. Is it hard to secure access to different food types? | 1 | |
| Healthcare | 4. Does anyone in your household use medication? | 1 | 5 |
| | 5. Does anyone in your household use five types of medications, or more, daily? | 1 | |
| | 6. Does anyone in your household have a health condition that requires continuous care? | 1 | |
| | 7. Is anyone in your household impaired to perform daily activities? | 1 | |
| | 8. Does anyone in your household need help accomplishing daily healthcare procedures? | 1 | |
| Family | 9. Did anyone in your household have an absent mother in childhood? | 1 | 3 |
| | 10. Did anyone in your household have an absent father in childhood? | 1 | |
| | 11. Has anyone in your household been abandoned by the family? | 1 | |
| Violence | 12. Does anyone in the household coexist with violent individuals? | 1 | 3 |
| | 13. Has anyone in your household been the victim of violence? | 1 | |
| | 14. Is there any violence happening in your home? | 1 | |
| **Total** | | **14** | **14** |

The final version for application of the EVFAM-BR is presented as S1 File (in English) and S2 File (in Portuguese).

## Discussion

The Family Vulnerability Scale for Brazil (EVFAM-BR) demonstrated evidence of content validity and internal structure in a multi-regional and multi-professional context, allowing for the measurement of family vulnerability in Brazil.

The EVFAM-BR consists of 14 items distributed across the dimensions: Income, Healthcare, Family, and Violence. It is worth noting that the EVFAM-BR aims to measure social vulnerability within the family context, and as such, all items refer to the family nucleus, rather than being directed to a specific household member or respondent.

**Table 12. Family Vulnerability Scale for Brazil (*Escala de Vulnerabilidade Familiar*, EVFAM-BR, in Portuguese).**

| Dimensão | Item | Escore Item | Escore Dimensão |
|---|---|---|---|
| Renda | 1. Alguém do domicílio passa por dificuldades financeiras? | 1 | 3 |
| | 2. Falta dinheiro para atender as necessidades do domicílio? | 1 | |
| | 3. Existem dificuldades de acesso a diferentes tipos de alimentos? | 1 | |
| Cuidado em Saúde | 4. Alguém no domicílio faz uso de medicamentos? | 1 | 5 |
| | 5. Alguém no domicílio faz uso de 5 ou mais tipos de medicamentos por dia? | 1 | |
| | 6. Alguém no domicílio possui condição de saúde que requer cuidados contínuos? | 1 | |
| | 7. Alguém no domicílio tem dificuldades para realizar atividades do dia a dia? | 1 | |
| | 8. Alguém no domicílio necessita de ajuda para realizar seus cuidados diários de saúde? | 1 | |
| Família | 9. Alguém no domicílio teve a mãe ausente durante a infância? | 1 | 3 |
| | 10. Alguém no domicílio teve o pai ausente durante a infância? | 1 | |
| | 11. Algum familiar já esteve em situação de abandono pela família? | 1 | |
| Violência | 12. Alguém no domicílio convive com pessoas violentas? | 1 | 3 |
| | 13. Alguém em seu domicílio já foi vítima de violência? | 1 | |
| | 14. Acontece violência em sua casa? | 1 | |
| **Total** | | **14** | **14** |

Next is an example of the EVFAM-BR applied to a family living in a given local and time: one of the household residents in a given house is facing financial difficulties. However, lack of money to meet household needs is not identified as an issue and there is no difficulties in having access to different food types; one of the household residents has a chronic disease that requires continuous care and needs controlled medication (less than five medication types daily), but none of the residents has any difficulties in performing daily activities and does not need help to perform daily healthcare procedures; none of the residents had either an absent mother or father during childhood and no family member was abandoned by the family; there was no violence in the house and no one in the house was coexisting with violent people, but one of the household residents was the victim of violence. Given the positive answers to items related to financial difficulties by one of the household residents (1), a health condition requiring continuous care (1), use of medication (1) and a person who was the victim of violence (1), this family would reach score 4, which according to the EVFAM-BR it represents a family classified as "low family vulnerability".

The literature consistently states that income is a relevant social health determinant that must be taken into account to allow planning an equitable healthcare provision [66]. More recent discussions about this topic led to different views on how income inequality affects health. Rich or poor individuals, in societies with low social cohesion, may become susceptible to a range of issues, including crime, insufficient public investments, and the adoption of unhealthy behaviors like smoking, excessive alcohol consumption, and sedentary lifestyles [67]. These outcomes can help us better understand the relationship between income-linked variables, facilitating interventions at macroeconomic level, and evaluating shifts in population health.

The healthcare dimension is timely, given the rapid aging of the population in Brazil and worldwide, a fact that demands reorganization of healthcare services in order to continuously meet population needs, in an organized way and based on quality and safety. With respect to the family dimension, several studies have shown the association between the absence of parents and a number of health outcomes, including cognitive development loss [68,69], impact on mental health [70,71], and early development of a health-related risk behaviors such as smoking and alcohol abuse [72].

Finally, the dimension "violence" reaffirms the observation outlined in the WHO's 2030 agenda for the sustainable development of millennium goals. This agenda highlights violence prevention as a pivotal component for fostering global development and enhancing overall quality of life. It is known that violence has a significant impact on health, extending beyond the initial trauma, exacerbating the likelihood of other significant causes of illness and death [73]. An association among exposure to violence, undesired health outcomes and unhealthy behaviors, such as drug and alcohol abuse, mainly among low-income mothers in urban locations, can be identified [74].

While there is a recognized urgency for research in low and middle-income countries, where 90% of global violence occurs, only 10% of studies on violence are conducted in these contexts [73]. Thus, EVFAM-BR emerges as a tool that enables the structured and routine introduction of social factors known to be associated with the health-disease process in the healthcare services in developing countries.

With the evidence of EVFAM-BR validity and the three proposed strata of family vulnerability resulting from its application, there is potential to contribute to the role of Primary Health Care (PHC) in executing its attributes, particularly in coordinating care from population-based management in the community and family context [75].

It is essential to highlight that the EVFAM-BR is a robust and concise instrument that initially demands a low investment of professional workload and financial resources. Because it is

an objective instrument (only "yes" and "no" answers are expected), all PHC professionals must be trained to use it. This instrument emerges as a powerful tool to support the work by community health agents, since these actors are community members and are closely bond to families in the territory; this process makes the "interview environment" more comfortable and trustful for users who answer the questions on behalf of the household. Therefore, EVFAM-BR can be implemented using both printed materials and online platforms, as it serves as a valuable tool integrated into the routine of healthcare teams. Also, there is potential to incorporate EVFAM-BR into the digital registration system of the Brazilian Unified Health System (SUS).

It is possible to prepare the teams at the time to plan their actions and health interventions (based on exposing families to conditions that increase vulnerability and risk to develop illnesses) by applying EVFAM-BR and by interpreting its household-classification results based on the three predicted strata. It is crucial to consider that vulnerability contexts can change overtime, so the instrument periodic application is necessary to keep teams planning updated according to the population's needs.

Among the limitations of this study is the convenience sampling, which does not allow for statistical representativeness of the results. Nevertheless, the study was conducted in various socioeconomic, demographic, and cultural contexts, including participants from all five Brazilian geographical regions. Additionally, the possibility of memory bias should be mentioned considering that participants answered questions regarding retrospective information on all residents of the household.

As strengths, data collection by external professionals trained to conduct the interviews and the use of robust techniques for identifying evidence of EVFAM-BR validity are noteworthy. One such technique is the Content Validity Ratio (CVR), which is a sophisticated method [76] if compared to other proposed alternatives [77]. CVR calculation takes into consideration the number of judges [24,25] and it minimizes the inflation of chance agreement [24], a fact that allows having more judges to assess the instrument. The adoption of a multi-disciplinary panel of judges comprising of researchers, translators, health professionals, methodology experts and lay people ensures better consistency of the results obtained [77–79].

It is important to highlight the need to implement research to identify the potentialities and challenges for the practical use of EVFAM-BR in the routine of PHC services in different Brazilian contexts, as well in countries with similar health systems, considering the socioeconomic, demographic, sanitary, and epidemiological profiles of populations.

## Conclusion

Given the techniques here applied and the results obtained, it is possible stating that the set of content validity and EVFAM-BR internal structure are adequate, consistent, reliable and robust. Also, the cross-validation method ensured model reliability and replicability. Here we present a concise scale capable of accurately measuring and distinguishing family vulnerability.

## Supporting information

**S1 File. Family Vulnerability Scale for Brazil (EVFAM-BR).**
(DOCX)

**S2 File. Family Vulnerability Scale (*Escala de Vulnerabilidade Familiar*, EVFAM-BR, in Portuguese).**
(DOCX)

**S3 File. EVFAM-BR database.**
(XLS)

## Author Contributions

**Conceptualization:** Evelyn Lima de Souza, Ilana Eshriqui, Flávio Rebustini, Daiana Bonfim.

**Data curation:** Evelyn Lima de Souza, Ilana Eshriqui, Flávio Rebustini, Ricardo Macedo Lima, Daiana Bonfim.

**Formal analysis:** Evelyn Lima de Souza, Ilana Eshriqui, Flávio Rebustini, Eliana Tiemi Masuda, Francisco Timbó de Paiva Neto, Ricardo Macedo Lima, Daiana Bonfim.

**Investigation:** Evelyn Lima de Souza, Ilana Eshriqui, Flávio Rebustini, Eliana Tiemi Masuda, Francisco Timbó de Paiva Neto, Ricardo Macedo Lima, Daiana Bonfim.

**Methodology:** Evelyn Lima de Souza, Ilana Eshriqui, Flávio Rebustini, Daiana Bonfim.

**Project administration:** Daiana Bonfim.

**Software:** Flávio Rebustini.

**Supervision:** Ilana Eshriqui, Flávio Rebustini, Daiana Bonfim.

**Validation:** Evelyn Lima de Souza, Ilana Eshriqui, Flávio Rebustini, Eliana Tiemi Masuda, Francisco Timbó de Paiva Neto, Ricardo Macedo Lima, Daiana Bonfim.

**Visualization:** Evelyn Lima de Souza, Ilana Eshriqui, Flávio Rebustini, Eliana Tiemi Masuda, Francisco Timbó de Paiva Neto, Ricardo Macedo Lima, Daiana Bonfim.

**Writing – original draft:** Evelyn Lima de Souza, Ilana Eshriqui, Flávio Rebustini, Eliana Tiemi Masuda, Francisco Timbó de Paiva Neto, Ricardo Macedo Lima, Daiana Bonfim.

**Writing – review & editing:** Evelyn Lima de Souza, Ilana Eshriqui, Flávio Rebustini, Eliana Tiemi Masuda, Francisco Timbó de Paiva Neto, Ricardo Macedo Lima, Daiana Bonfim.

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
