## [Decision Letter · Decision Letter 0]

21 Jun 2023

PONE-D-22-35161

Family Vulnerability Scale: evidence of content and internal structure validity

PLOS ONE

Dear Dr. Souza,

Thank you for submitting your manuscript to PLOS ONE. After careful consideration, we feel that it has merit but does not fully meet PLOS ONE’s publication criteria as it currently stands. Therefore, we invite you to submit a revised version of the manuscript that addresses the points raised during the review process.

We look forward to receiving your revised manuscript.

Kind regards,

Isabel Cristina Gonçalves Leite

Academic Editor

PLOS ONE

2. In the ethics statement in the Methods, you have specified that verbal consent was obtained. Please provide additional details regarding how this consent was documented and witnessed, and state whether this was approved by the IRB.

Additional Editor Comments:

The interesting contribution given to the literature by the presented manuscript can be improved according to the comments of its reviewers. We are awaiting your return

Reviewers' comments:

Reviewer's Responses to Questions

**Comments to the Author**

1. Is the manuscript technically sound, and do the data support the conclusions?

Reviewer #1: Yes

Reviewer #2: Yes

2. Has the statistical analysis been performed appropriately and rigorously? 

Reviewer #1: Yes

Reviewer #2: Yes

3. Have the authors made all data underlying the findings in their manuscript fully available?

Reviewer #1: Yes

Reviewer #2: Yes

4. Is the manuscript presented in an intelligible fashion and written in standard English?

Reviewer #1: No

Reviewer #2: Yes

5. Review Comments to the Author

Reviewer #1: The study is relevant and well structured, but still needs to undergo English revision.

The abstract must have an explicit presentation of the objective.

In Brazil, who are the professionals who make up primary health care? Describe in the text, because the results show some professionals that we do not know if they are all or part of them.

What are the criteria to define professionals able to participate?

What were the exclusion criteria defined by the authors?

The criteria established for exclusion of the participant were not clear.

What are the exclusion criteria for service users to participate in the survey? As the answers were via Radcap, could everyone access the internet using a smartphone or other technology?

The discussion focused on describing the results and explaining the possibilities of the Brazilian version.

Note a lack of comparative discussion with the literature, in addition to comparison with the original instrument and/or other validated versions of it.

It is necessary to inform the limitations of the study.

Reviewer #2: The study developed a new instrument to investigate the social vulnerability of families in the context of PHC and describes the evaluation of its content and internal structure validity. The topic is very important, especially in this post-pandemic moment, when the social situation of families has deteriorated significantly. A more comprehensive instrument was really lacking, addressing the issue in a multidimensional way and having its validity evaluated in depth. The study was well conducted, the methodological path is well described and the description of the results and their interpretation are consistent with what was presented in the tables. The effort to have included participants from all regions of Brazil in its various stages deserves praise, which contributed to adapting the instrument to the various Brazilian regional realities. The instrument will be very useful in investigating the social vulnerability of families in Brazil and in other socioeconomically similar countries.

6. PLOS authors have the option to publish the peer review history of their article (what does this mean?). If published, this will include your full peer review and any attached files.

Reviewer #1: No

Reviewer #2: No

---

## [Author Response · Author response to Decision Letter 0]

6 Aug 2023

Response to Reviewers

São Paulo, July 27th, 2023

Dear Academic Editor and Reviewers,

We are grateful for the valuable review of our work. Such thoughtful suggestions and comments have help us to improve our manuscript substantially. Please find below our point-by-point responses.

Kind regards,

Evelyn Lima de Souza on behalf of all authors

Answer: The revision meets PLOS ONE’s style requirements.

2. In the ethics statement in the Methods, you have specified that verbal consent was obtained. Please provide additional details regarding how this consent was documented and witnessed, and state whether this was approved by the IRB.

Answer: In accordance, we have clarified the following information in the Methods (p. 8):

“PHC users over 18-years-old were invited to participate. The Informed Consent Form approved by Hospital Israelita Albert Einstein in Ethics Research Committee was applied by the data collector. Verbal acceptance or refusal was registered using the RedCap® software on a tablet device and the participant received a printed copy of the ICF. Next, the structured questionnaire about the family vulnerability scale was applied and registered offline using the RedCap® installed on a tablet device. At the end of each day, information registered on each tablet was uploaded to the RedCap® server.”

Answer: The Supporting Information and in-text citations have been updated (p. 34): 

“Table 11 (in English) and Table 12 (in Portuguese) summarize the Family Vulnerability Scale, with their respective dimensions, items, and scores.

The final version for application of the EVFAM-BR is presented as S1 Supporting Information (in English) and S2 Supporting Information (in Portuguese).”

Answer: The reference list has been revised. The web address to reference number 8 has been corrected due to unavailability of the previous web page and citation date has been updated.

Additional Editor Comments:

The interesting contribution given to the literature by the presented manuscript can be improved according to the comments of its reviewers. We are awaiting your return

Reviewers' comments:

Reviewer's Responses to Questions

Comments to the Author

1. Is the manuscript technically sound, and do the data support the conclusions?

Reviewer #1: Yes

Reviewer #2: Yes

2. Has the statistical analysis been performed appropriately and rigorously?

Reviewer #1: Yes

Reviewer #2: Yes

3. Have the authors made all data underlying the findings in their manuscript fully available?

Reviewer #1: Yes

Reviewer #2: Yes

 4. Is the manuscript presented in an intelligible fashion and written in standard English?

Reviewer #1: No

Reviewer #2: Yes

Answer: We thank reviewers’ attention and inform that the language has been reviewed and improved.

Review Comments to the Author

Reviewer #1: The study is relevant and well structured, but still needs to undergo English revision.

Answer: We thank reviewers’ attention and inform that the language has been reviewed and improved.

The abstract must have an explicit presentation of the objective.

Answer: We appreciate this remark. The aim of the study is to develop and search for evidences about the validity of the EVFAM-BR. We performed a brief adjustment in the abstract to clarify the objective of the study (p. 2): 

“The primary objective of this study is to develop and gather evidence on the validity of the Family Vulnerability Scale for Brazil, commonly referred to as EVFAM-BR (in Portuguese).”

In Brazil, who are the professionals who make up primary health care? Describe in the text, because the results show some professionals that we do not know if they are all or part of them.

Answer: We have clarified who are some of the primary health care professionals in Brazil who participated in our study (p. 5): 

“PHC professionals (such as physicians, nurses, nursing assistants, community health agents, oral health team and multi-professional team, as psychologists, social workers, speech therapists, physiotherapists, nutritionists, and pharmacists) from all geographic regions in Brazil were invited to join the first qualitative exploratory stage of the study to define the concept of “family vulnerability” and to identify factors likely associated with it”

What are the criteria to define professionals able to participate? What were the exclusion criteria defined by the authors?

The criteria established for exclusion of the participant were not clear.

What are the exclusion criteria for service users to participate in the survey? 

Answer: We thank the reviewers’ attention and inform that these information are now detailed in the manuscript.

“PHC professionals (such as physicians, nurses, nursing assistants, community health agents, oral health team and multi-professional team, as psychologists, social workers, speech therapists, physiotherapists, nutritionists, and pharmacists) from all geographic regions in Brazil were invited to join the first qualitative exploratory stage of the study to define the concept of “family vulnerability” and to identify factors likely associated with it. The insights gathered from this stage were used to inform the development of items for the instrument. This effort was done to identify different views of family vulnerability in different geographic regions countrywide.

The invitation was conducted using a snowball sampling method [17], through an online questionnaire on the RedCap® electronic tool [18, 19] sent via WhatsApp and email. After reading the Informed Consent Form (ICF) and providing consent to participate in the study, participants were given access to a semi-structured questionnaire.” (p. 5, 6).

“The panel encompassed health professionals, scholars and psychometrists who were invited to join the study through the snowball method and like previously, an online RedCap® questionnaire with ICF was used. 

Registered participants accepted to participate and signed an ICF, but did not access the questionnaires related to the family vulnerability concept in the first qualitative exploratory stage or the questionnaire used to evaluate the items in the initial version of the instrument during the panel of judges were excluded from the study” (p. 6).

“PHC users over 18-years-old were invited to participate. The Informed Consent Form approved by Hospital Israelita Albert Einstein in Ethics Research Committee was applied by the data collector. Verbal acceptance or refusal was registered using the RedCap® software on a tablet device and the participant received a printed copy of the ICF. Next, the structured questionnaire about the family vulnerability scale was applied and registered offline using the RedCap® installed on a tablet device. At the end of each day, information registered on each tablet was uploaded to the RedCap® server. 

PHC users who accepted to participate according to the ICF but had missing information regarding the family vulnerability scale were excluded.” (p. 8, 9).

As the answers were via Radcap, could everyone access the internet using a smartphone or other technology?

Answer: Participants of the exploratory stage and panel of judges were able to access and answer the RedCap questionnaire using any device connected to the internet. PHC users responded to the survey conducted by the data collector, who recorded the answers offline using RedCap installed on a tablet belonging to the research team. To clarify this process, brief adjustments have been made to the Methods section:

“The panel encompassed health professionals, scholars and psychometrists who were invited to join the study through the snowball method and like previously, an online RedCap® questionnaire with ICF was used. 

Registered participants accepted to participate and signed an ICF, but did not access the questionnaires related to the family vulnerability concept in the first qualitative exploratory stage or the questionnaire used to evaluate the items in the initial version of the instrument during the panel of judges were excluded from the study” (p. 6).

“Next, the structured questionnaire about the family vulnerability scale was applied and registered offline using the RedCap® installed on a tablet device. At the end of each day, information registered on each tablet was uploaded to the RedCap® server.” (p. 8).

The discussion focused on describing the results and explaining the possibilities of the Brazilian version.

Note a lack of comparative discussion with the literature, in addition to comparison with the original instrument and/or other validated versions of it.

Answer: We thank the reviewer comment, but this is an unprecedented study that considers a wide concept of family vulnerability, according to PHC professionals during the exploratory stage. Further, our study presents a scale for measuring family vulnerability with robust evidence of validity nationwide, considering different Brazilian settings. Previous instruments, mentioned at reference number 11 to 15, are limited to information routinely registered in PHC records and/or were validated in a specific context. Thus, we mentioned these previous instruments at the introduction section, but it was not possible to compare them with our results, so we opted to focus our discussion on the dimensions considered to measure the vulnerability of families by the EVFAM-BR.

It is necessary to inform the limitations of the study.

Answer: We thank the reviewer attention and inform that we clarify the limitations of the study at the Discussion section (p. 40):

“Among the limitations of this study is the convenience sampling, which does not allow for statistical representativeness of the results. Nevertheless, the study was conducted in various socioeconomic, demographic, and cultural contexts, including participants from all five Brazilian geographical regions. Additionally, the possibility of memory bias should be mentioned considering that participants answered questions regarding retrospective information on all residents of the household.”

Reviewer #2: The study developed a new instrument to investigate the social vulnerability of families in the context of PHC and describes the evaluation of its content and internal structure validity. The topic is very important, especially in this post-pandemic moment, when the social situation of families has deteriorated significantly. A more comprehensive instrument was really lacking, addressing the issue in a multidimensional way and having its validity evaluated in depth. The study was well conducted, the methodological path is well described and the description of the results and their interpretation are consistent with what was presented in the tables. The effort to have included participants from all regions of Brazil in its various stages deserves praise, which contributed to adapting the instrument to the various Brazilian regional realities. The instrument will be very useful in investigating the social vulnerability of families in Brazil and in other socioeconomically similar countries.

Answer: We thank the reviewer attention and comment.

6. PLOS authors have the option to publish the peer review history of their article (what does this mean?). If published, this will include your full peer review and any attached files.

Do you want your identity to be public for this peer review? For information about this choice, including consent withdrawal, please see our Privacy Policy.

Reviewer #1: No

Reviewer #2: No

---

## [Editor Report · Decision Letter 1]

12 Sep 2023

Family Vulnerability Scale: evidence of content and internal structure validity

PONE-D-22-35161R1

Dear Dr. Souza,

We’re pleased to inform you that your manuscript has been judged scientifically suitable for publication and will be formally accepted for publication once it meets all outstanding technical requirements.

Kind regards,

Isabel Cristina Gonçalves Leite

Academic Editor

PLOS ONE

Additional Editor Comments (optional):

The authors followed the recommendations made by the reviewers

---

## [Editor Report · Acceptance letter]

18 Sep 2023

PONE-D-22-35161R1 

Family Vulnerability Scale: evidence of content and internal structure validity 

Dear Dr. Souza:

I'm pleased to inform you that your manuscript has been deemed suitable for publication in PLOS ONE. Congratulations! Your manuscript is now with our production department. 

Kind regards, 

on behalf of

Dr. Isabel Cristina Gonçalves Leite 

Academic Editor

PLOS ONE